

# Assessment of Sun photometer Langley calibration at the high-elevation sites Mauna Loa and Izaña

Carlos Toledano[1], Ramiro González[1], David Fuertes[1,2], Emilio Cuevas[3], Thomas F. Eck[4,5], Stelios Kazadzis[6], Natalia Kouremeti[6], Julian Gröbner[6], Philippe Goloub[7], Luc Blarel[7], Roberto Román[1], África Barreto[8,3,1], Brent N. Holben[4], and Victoria E. Cachorro[1]

[1]Group of Atmospheric Optics, University of Valladolid (GOA-UVa), Spain
[2]GRASP-SAS, Lille, France
[3]Izaña Atmospheric Research Center, Meteorological State Agency of Spain (AEMET), Tenerife, Spain
[4]NASA Goddard Space Flight Center, Greenbelt, MD, USA
[5]Universities Space Research Association, Columbia, MD, USA
[6]Physikalisch-Meteorologisches Observatorium Davos, World Radiation Center – PMOD/WRC, Davos, Switzerland
[7]Laboratory of Atmospheric Optics, University of Lille, Villeneuve d'Ascq, France
[8]Cimel Electronique, Paris, France

*Correspondence to:* Carlos Toledano (toledano@goa.uva.es)

**Abstract.** The aim of this paper is to analyze the suitability of the high-mountain stations Mauna Loa and Izaña for Langley plot calibration of Sun photometers. Thus the aerosol optical depth (AOD) characteristics and seasonality, as well as the cloudiness, have been investigated in order to provide a robust estimation of the calibration accuracy, as well as the number of days that are suitable for Langley calibrations. The data used for the investigations belong to AERONET and GAW-PFR
networks, which maintain reference Sun photometers at these stations with long measurement records: 22 years at Mauna Loa and 15 years at Izaña. In terms of clear sky and stable aerosol conditions, Mauna Loa (3397m a.s.l.) exhibits on average of 377 Langleys (243 morning and 134 afternoon) per year suitable for Langley plot calibration, whereas Izaña (2373m a.s.l.) shows 343 Langleys (187 morning and 155 afternoon) per year. The background AOD(500nm) values, on days that are favorable for Langley calibrations, are in the range 0.01-0.02 throughout the year, with well defined seasonality that exhibits a spring
maximum at both stations plus a slight summer increase at Izaña. The statistical analysis of the long-term determination of extraterrestrial signals yields to a calibration uncertainty of ~0.2-0.5%, being this uncertainty smaller in the near infrared and larger in the ultraviolet wavelengths. This is due to atmospheric variability that cannot be reduced based only on quality criteria of individual Langely plots.

## 1 Introduction

The Langley plot method (Shaw, 1983) is widely used for absolute calibration of Sun photometers. The main requirement for the method to be successful is the atmospheric transmittance stability during the period in which direct Sun observations are acquired at varying solar elevations. Apart from the original (classic) approach, several variations have been developed



(e.g. Herman et al., 1981; Forgan, 1994; Campanelli et al., 2004). These are mostly intended to reduce the uncertainty and calibration error in case of changes in the atmospheric transmittance during the observation period.

Sun photometer networks like the AErosol RObotic NETwork (AERONET, Holben et al., 1998), the Global Atmospheric Watch – Precision Filter Radiometer (GAW-PFR, Wehrli, 2005), Skyradiometer Network (SKYNET, Nakajima et al., 1996),

use the Langley plot method to calibrate the direct Sun channels, i.e. obtain extraterrestrial signals ($V_0$), with the aim of calculating aerosol optical depth (AOD). Although some networks (e.g. SKYNET) perform Langleys 'on site' (Campanelli et al., 2007), networks like AERONET and GAW only use high altitude stations to provide accurate absolute calibration with the Langley plot method in the so-called master instruments. The calibration is later transferred to field instruments by comparison in a calibration platform.

The AERONET network currently has 3 calibration centers: Goddard Space Flight Center (GSFC, in Greenbelt, Maryland), Laboratory of Atmospheric Optics (LOA, in Lille/Carpentras, France) and Group of Atmospheric Optics (GOA, in Valladolid, Spain). The GSFC master instruments are calibrated at the Mauna Loa Observatory, in Hawaii. The LOA and GOA masters are calibrated at Izaña Observatory. The GAW-PFR network is managed by the Physikalisch Meteorologisches Observatorium Davos, World Radiation Center (PMOD/WRC) at Davos (Switzerland). It uses a triad of reference (PFR) instruments at Davos

which are considered by the World Meteorological Organization (WMO-GAW) as the reference instrument triad for AOD measurements. It also operates permanent reference instruments at Izaña and at Mauna Loa, that return periodically (every six months) to PMOD/WRC and are compared with the reference triad (Kazadzis et al., 2018b).

Mauna Loa is a reference site for radiometric observations and calibrations. It was very early considered as an ideal place for calibration of Sun photometers using the Langley technique (Shaw, 1979), hence it hosts reference instruments of the main

radiometric networks. Many studies have already reported the atmospheric aerosol characteristics at Mauna Loa (Bodhaine et al., 1981, 1992; Dutton et al., 1994; Andrews et al., 2011; Hyslop et al., 2013), to cite some. Numerous studies about aerosol characteristics at Izaña have also been conducted (e.g. Prospero et al., 1995; Rodríguez et al., 2011; García et al., 2016). Izaña is also commonly used for accurate Langley plot calibrations (even in Moon photometry, Barreto et al. (2013, 2016)), although the site performance has not yet been quantitatively evaluated in this sense.

After years of continuous Sun photometer observations at the Mauna Loa and Izaña observatories, long and high quality measurement records are available, and the quantification of the calibration performance can be accomplished with the support of robust datasets. Therefore, the aim of this paper is to analyze the capability of the two high-mountain stations Mauna Loa and Izaña for Langley plot calibration, in terms of aerosol characteristics, seasonality and cloudiness; and provide statistically robust figures for calibration accuracy. The data used for the investigations belong to AERONET and GAW-PFR networks,

both having reference instruments at these stations with long measurement records.



## 2 Sites and instrumentation

### 2.1 The Mauna Loa and Izaña observatories

The atmospheric stability required for the Langley plot method is more easily achieved in remote, high-elevation locations, especially because the AOD is very low. Several characteristics make Izaña and Mauna Loa Observatories to be unique for this

purpose.

The Izaña Observatory (Tenerife, Spain, 28°N, 16°W) is located at the top of a mountain plateau, 2373 m above sea level, about 15 km away from the Teide peak. It is run by the Meteorological State Agency of Spain (AEMET, see http://izana.aemet.es). Izaña is normally above a strong temperature inversion layer and therefore free of local anthropogenic influence. It is a World Meteorological Organization (WMO) Global Atmospheric Watch (GAW) program station as well as WMO-CIMO Testbed

for Aerosols and Water Vapour Remote Sensing Instruments (http://testbed.aemet.es). It hosts reference instruments of several radiometric networks (e.g. Regional Brewer Calibration Centre, GAW-PFR, AERONET, PANDORA, etc.). Details of the Izaña facilities and activities are described in Cuevas et al. (2017b).

The Mauna Loa Observatory (Big Island, Hawaii, 19°N, 155°W) is located on the slope of Mauna Loa volcano, 3397 m above sea level. It was created in 1956 and run by the National Oceanic and Atmospheric Administration (NOAA, see

https://www.esrl.noaa.gov/gmd/obop/mlo). It is reference observatory for a wide set of atmospheric composition research programs (greenhouse gases, carbon cycle, aerosols, water vapor, ozone, trace gases, etc.) and has been continuously monitoring and collecting data related to the atmospheric change.

Both observatories are located in the free troposphere. The aerosol content above is very low (see section 3), as well as the water vapor column (PWV, precipitable water vapor) and the molecular (Rayleigh) optical depth. For instance the water

vapor column at Izaña ranges from 0.2cm in winter to 0.7cm in summer (monthly averages, AERONET-derived, see table S1) whereas in the nearby site 'Santa_Cruz_Tenerife' located at sea level, the PWV ranges from 1.5cm to 2.5cm. The atmosphere is therefore very stable, especially in the mornings. In the afternoon, local convection can rise the boundary layer up to the Observatory level, especially at Mauna Loa. The strong inversion associated to the Trade Wind at Izaña very often prevents from boundary layer to reach the observatory (Carrillo et al., 2015).

Another important feature to assure the success of the Langley calibration, is to reduce as much as possible the time needed to acquire Sun observations at a wide optical air mass range, in order to avoid possible atmospheric changes. The latitude of Mauna Loa and Izaña, close to the tropics, make the air mass to change rapidly from 7 to 2, i.e. solar elevations from 8° to 30° approximately, lasting about 1:35h to 2:15h depending on the season (the duration is few minutes shorter for Mauna Loa). Just for comparison, at 37° latitude, the time in winter to change from air mass 7 to 2 is more than 3h. At higher latitudes, air mass

2 is not reached in winter.

The cloudiness is another main aspect in performing Langleys. Even thin high clouds perturb the Langley calibration dramatically. To evaluate the sky conditions with the same methodology at both locations, a cloud satellite product has been used. In particular, the cloud products (GDP-4.8 version) of the algorithms OCRA and ROCCIN (Loyola R. et al., 2010) from GOME-2 onboard MetOp-A have been used to evaluate cloud fraction and cloud top height respectively. The cloud top height



is a crucial parameter due to the high elevation of the observatories. The monthly mean cloud fraction and number of clear sky days, defined as cloud fraction $< 0.1$, have been evaluated over the period 2007-2014. If the cloud top height was lower than the site elevation, the cloud fraction was considered 0. The results are shown in Table 1. On average, Mauna Loa exhibits 24 clear sky days per month, whereas Izaña has 20. There is some seasonal variability, being the period between May and August

the most sunny at both locations. However it is possible that very thin cirrus (optical depth $< 0.1$) are not detectable in these satellite products. This will be taken into account in the analysis of the Langley regressions (section 4).

Besides the necessary atmospheric conditions, the facility itself including permanent and trained staff, convenient access and easy logistics are also an important point to consider. Actually the capacity of the measurement platforms themselves is a limitation given that many radiometric networks have reference instruments in these two observatories. This limitation together

with the relatively expensive shipping to such remote locations, is the main reason for AERONET (and many other networks) to calibrate master instruments with the Langley method at Izaña and Mauna Loa, and then transfer the absolute calibration to field instruments in calibration platforms located in much more accessible facilities at GSFC, Carpentras, Davos, Valladolid, etc. As example, 15 to 20 calibrations of AERONET master instruments are accomplished every year at Izaña. Of course the calibration accuracy of the field instruments is therefore less than that of masters, but logistically it is not reasonable to ship

several hundred instruments every year to Mauna Loa or Izaña. The AOD calibration accuracy needed for field instruments (0.01 to 0.02 absolute error as recommended by Kazadzis (2016)) can be achieved by means of side-by-side inter-calibration (Holben et al., 1998; Eck et al., 1999). Possible instrument fluctuations due to shipping are controlled by using always a couple of masters that travel together and rigorous comparison of master instruments at the inter-calibration sites. Ratio of Sun direct signals between the two masters must keep below $1\%$ variability.

## 2.2   Instrumentation and datasets

The AERONET standard instrument is the Cimel-318, that has been extensively described (e.g. Holben et al., 1998). It is an automatic radiometer equipped with a 2-axis robot, that collects both direct Sun and sky radiance observations at selected wavelengths in the range 340 to 1640nm. Three generations of Cimels have been used in AERONET: the first (starting the early 1990's) were analog instruments. After 2002 the digital version (Cimel 318N) came into play, and after 2013 the so-

called Triple instruments (Cimel 318T, after Sun-Sky-Moon measurement capability) started to operate. All three types of instruments can still be found nowadays in AERONET.

The Precision Filter Radiometer of the GAW-PFR network is described in detail in Wehrli (2005). It uses four AOD channels at 368, 412, 500 and 862 nm and needs a separate solar tracker. It is designed for long-term stability, therefore the detectors are behind a shutter except for the brief sampling periods and the instrument is stabilized in temperature and hermetically sealed,

having internal atmosphere of pressurized dry nitrogen.

Both instruments use interference filters to select the wavelengths, with full width at half maximum of about 2-10nm (filters are narrower in the ultraviolet wavelengths). The PFR uses one optical path and detector per channel, allowing simultaneous (and continuous) observation in the 4 channels. Conversely, the Cimel has a single detector (or 2 in the case of instruments with 1640nm channel) and the filters are mounted in a rotating filter wheel. The Cimel configuration allows more wavelength



channels (up to 10) but they can only be measured sequentially. In automatic operation, the Cimel takes a triplet measurement (3 separate measurements in a 1-minute interval) every 15 minutes (or 3 minutes, in the high frequency sampling mode), although during the 'Langley sequence' –am or pm for air masses larger than 2– the Cimel measures at fixed solar elevations, with higher frequency.

The AERONET observations at Mauna Loa started in 1994. The observation period used in this study spans 20 years (1997-2016). Within this period, 210 deployments of 22 different master photometers were done. This gives an idea of the frequent swap of Cimel instruments, once per month on average. The AERONET measurements at Izaña started in 2003 and had 37 deployments (71 days on average, 16 different instruments) until January 2011, when instrument #244 was set as permanent reference. The GAW-PFR measurements started in 2000 and 2001 at Mauna Loa and Izaña respectively. The list of PFR

radiometers deployed at each location is given in table 2. The high long-term stability of these radiometers will be shown in section 4.

The database tool 'CÆLIS' (Fuertes et al., 2017, www.caelis.uva.es), developed at the Group of Atmospheric Optics, University of Valladolid (GOA-UVa) since 2008, has been used to facilitate the organization and extraction of data. It consists of a relational database, a web interface and a real-time data processing module. The 'demonstrat' software tool (Holben et al.,

1998) was used to browse the AERONET data and construct the AERONET data sets at the two stations, given the frequent swap out of master instruments (every 3-4 months). Conversely the GAW-PFR data sets are composed by few instruments deployed for very long periods.

The two approaches have been therefore different, being AERONET priority to frequently recalibrate and maintain the master instruments, shipping them to the inter-calibration platforms, whereas GAW-PFR has prioritized the stability in the

long-term observations, in order to facilitate the assessment of trends in the aerosol content, well in line with the GAW aims. However in the last years (since 2011) AERONET has a permanent instrument at Izaña, not involved in the rotation of masters between this site and the inter-calibration platforms.

## 3   Aerosol Climatology

The aerosol characteristics at Mauna Loa and Izaña observatories can be well established thanks to the long records of the

AERONET and GAW-PFR networks. The very low aerosol optical depth is a general feature at Mauna Loa throughout the year. At Izaña, very clean days alternate with some desert dust intrusions, especially in spring and summer (Cuevas et al., 2017a). The overall statistics for aerosol optical depth at 500nm wavelength is provided in Figure 1 and Table S1. These are computed by averaging all available daily mean values in the investigated period within a certain month of the year. As indicated above, 20 years of continuous AERONET data are used for Mauna Loa and 13 years for Izaña. Version 2 AERONET

data have been used in this analysis. As for GAW-PFR data, 15 years are available at Mauna Loa and 14 years at Izaña. Both are depicted in Figure 1. Although the measurement periods are different, the long-term averages of AERONET and GAW-PFR differ less than 0.01 for all months, with mean absolute difference of 0.0035 for the monthly means. This difference also fulfills the WMO criterion for intercomparison (WMO, 2005), which is set to $0.005 + 0.010/airmass$ (Kazadzis et al., 2018a).



The cloud screening methodologies of AERONET and GAW differ, thus contributing to differences in monthly means. AERONET uses the algorithm by Smimov et al. (2000), based on temporal variance as utilized by AERONET. GAW data are cloud screened following the methodology by Wehrli (2008). Other authors have accomplished extensive comparison of Cimel and PFR observations (Kim et al., 2008; Kazadzis et al., 2014, 2016, 2018a) with excellent results.

Regarding Mauna Loa (Fig. 1), the AOD (500nm) has a mean value of 0.016 (geometric mean 0.013), peaks in March with 0.028 and is minimum in August-September, with 0.011. The AOD (500nm) daily mean only exceeded 0.05 in 0.6% of the days. The monthly standard deviations indicate that the variability within each month is very low too. The largest variability is found from March through May, with monthly standard deviations about 0.015. The Ångström exponent AE(440-870nm), also given in Table S1, shows a mean value of 1.25, that is indicative of dominance by fine mode particles. The AE is slightly lower

in May (1.02), indicating somewhat greater proportion of coarse mode particles. The spring peak in aerosol concentration at Mauna Loa is a well documented phenomenon and it is attributed to the advection of Asian dust (e.g. Bodhaine et al., 1981; Perry et al., 1999). The uncertainty in AE is very high at MLO since the uncertainty in AOD (about $0.002 - 0.003$) is quite large in relation to the $\sim 0.01$ measured AOD. Thus the AE values at MLO should be in general taken with caution.

The low AOD makes it difficult to investigate any other aerosol optical and microphysical properties, in particular those

derived from the inversion of sky radiances for the AERONET instruments using the Dubovik inversion code (Dubovik and King, 2000; Dubovik et al., 2006). Such properties, like single scattering albedo or complex refractive index, are generally not quality assured if AOD(440nm) is less than 0.4 (Holben et al., 2006). Given that the AOD stability is the main requirement for Langley calibrations, in-depth investigation of the aerosol properties is not in the scope of this work and will not be considered here.

The mean AOD (500nm) at Izaña Observatory is 0.054 (geometric mean 0.029), with important seasonal variability. The difference between arithmetic and geometric mean is a good indicator of log-normal distribution of the AOD data (O'Neill et al., 2000). Monthly means range from 0.02 –November through February– up to 0.14 in July and August (geometric means 0.07 on both months, see Fig. 1). The transport of Saharan dust over Izaña in Summer enhances the aerosol content and the variability, as indicated by the large monthly standard deviations up to 0.15 in July. The Ångström exponent, that has a mean

value of 0.99, exhibits a clear decrease in the summer months down to 0.54 in August, confirming the predominance of coarse dust particles. Despite this variability, 25th percentile of AOD is $< 0.03$ in July and August, indicating a relevant portion of pristine days during the summer months.

In order to assess the dust event frequency over Izaña, the presence of dust has been investigated within the 13-year AERONET database. Following similar methodology that proposed by Toledano et al. (2007), dust events were identified

by daily mean $AOD(870nm) > 0.05$ and $AE < 0.6$, which approximately correspond to the 75th and 25th percentiles of these magnitudes in the Izaña dataset. This simple approach results in the identification of 58 dust event days per year on average. The seasonal distribution is not even. On the contrary, dust events are very rare from October to February (1-2 days per month), while July and August, on average, exhibit 16 and 17 dust event days respectively, which cause the higher AOD values observed in these months (Fig. 1). Similar results, even with slightly different methodology, were achieved by Guirado-Fuentes

et al. (2017).





The dust occurrence over Izaña in summer may yield to the incorrect conclusion that, during several months each year, the Langley calibrations are not possible in this station. But as it was previously indicated, dust events alternate with very clean (background) conditions. To demonstrate this important feature, all daily means of AOD (440nm) over 2004-2014 have been plotted as a function of the day of the year (Figure 2b). For comparison, Figure 2a displays the same plot for Mauna Loa.

As can be seen, most of the daily observations at Izaña (about 75%) correspond to background values. Higher daily means, corresponding to dust events, are evident from June to September. Dust events are less frequent and with lower AOD outside those months. Note that dust transport in winter occurs at much lower altitude than in summer, therefore the aerosol column above the observatory is minor in winter as compared to summer dust events, in which dust can reach 5 km height (Ansmann et al., 2011; Guirado-Fuentes, 2015; Cuevas et al., 2015). Izaña is therefore a privileged location for studying Saharan dust

within the Saharan Air Layer.

Another feature of the AOD seasonal cycle is the increase of the background AOD (lowest values) from March to May, with maximum background of about day of the year equal to 120, i.e. beginning of May. This is not exactly in phase with the spring AOD peak at Mauna Loa (in April). The background AOD is in May about 0.016 (440nm), whereas the rest of the year it is as low as AOD=0.005. Interestingly, this enhanced background occurs both at Mauna Loa and Izaña (Figure 2), although it is

15 unclear whether these two seasonal maxima have the same origin.

## 4  Assessment of calibration capability

### 4.1  Langely plot analysis

In order to investigate the station capability for Langley calibration, a software tool has been developed and integrated in CÆLIS (Fuertes et al., 2017). It performs two Langley plots for each available day (morning and afternoon, i.e. 'am' and

20 'pm') and stores the resulting extraterrestrial signal together with a set of regression statistics: correlation coefficient, standard deviation of the fit ($\sigma$), number of valid points, air mass range, fitting error for slope and intercept, etc. The routine performs the linear fit from airmass 7 to 2[1], and analyzes the standard deviation of the fit. If the residual for some point is larger than $2\sigma$, the point is eliminated and a new iteration starts until all points are within $2\sigma$ or the number of remaining points is less than 10. If $\sigma > 0.2$ or there is not enough number of points, the process stops.

This type of automatic and iterative analysis, allows identifying whether a certain day is suitable for Langley plot calibration according to pre-established quality thresholds. In our study, we have considered that for a certain period (morning or afternoon) within a particular day, the Langley calibration is possible if $\sigma < 0.006$, the number of valid points is $> 33\%$ of the initial number of observations (Harrison and Michalsky, 1994) and $AOD(500nm) < 0.025$. These criteria can be chosen based on experience (Kiedron and Michalsky, 2016), but they are not critical in this study because we do not intend to perform the

calibration of any particular instrument. For instance, for calibration purposes a higher threshold in $\sigma$ should be used for the UV wavelengths. However our purpose here is to analyze the number of suitable Langley plots in a climatological sense. Other

---

[1]This differs from the airmass range used in AERONET for Langley calibrations, i.e. 5 to 2, and 4 to 2 for the two UV channels (380 and 340 nm), thereby avoiding errors in optical airmass determination that increase significantly at larger airmass (Russell et al., 1993).





statistical indicators of the linear regression quality, such as the correlation coefficient, do not have enough sensitivity to be used for this purpose.

It is then straightforward to search the database for Langley periods fulfilling the indicated criteria. The results are given in Figure 3, in which the average number of Langley plots for each month is indicated, as well as the standard deviation resulting from the year-to-year variability. Morning and afternoon Langleys are given separately. It is common practice to use only mornings for Langley calibration, but in principle both periods are possible and therefore will be both considered in our study. Overall, Mauna Loa meets the selected criteria in 377 Langleys per year (243 'am' calibrations and 134 'pm'). This means, on average, about 20 morning Langleys and 11 afternoon Langleys per month. Izaña meets the criteria in 343 Langleys per year (187 'am' calibrations and 155 'pm'), which means 15 morning and 13 afternoon Langleys per month. There is certain seasonality, with less suitable days in spring and fall at Mauna Loa and better conditions from May through September and December-January. At Izaña the dust events reduce the number of suitable days in July-August, and the best time of the year is May-June.

The AOD (500nm) for the selected 'Langley' days, is given in Fig. 1b, in which monthly averages are calculated for comparison with the overall climatology (Fig. 1a). This plot provides the seasonality of the background AOD values, that exhibits a spring maximum at both stations plus a slight summer increase at Izaña.

## 4.2 Calibration and statistical uncertainty

A major issue pointed out by many authors is that, despite the available Langley plots can be screened with very strict criteria, a certain variability, i.e. uncertainty in the extraterrestrial signals, remains (Kazadzis, 2016). The noise is caused by small changes in atmospheric transmission having a hyperbolic (solar air mass) dependence, thus they do not affect the linearity of the Langley plot but may change the result (Shaw, 1976; Cachorro et al., 2004). That is also the reason not to use the correlation coefficient to discriminate Langley plots. This noise is well illustrated in Figure 4, in which the GAW-PFR data at Mauna Loa have been selected. They are very appropriate for this analysis due to the long deployment periods. We can see the daily extraterrestrial signals (500nm) obtained with the Langley plot method, after screening with the above mentioned criteria. Making the criteria even stricter reduces of course the number of available points, but does not reduce the variability much farther. That is the reason why many authors propose (and it is common practice) averaging a sufficient number of Langley plots to be able to achieve a satisfactory calibration (Slusser et al., 2000; Kazadzis, 2016).

For long deployments, such as the PFR's in Figure 4, the temporal fit to the extraterrestrial signals $V_0$ resulting from the Langley plots is better than just averaging, because it will take into account slow degradation of the optical elements (filters, detectors), which is quite clear, although small, in the plot. For instance, PFR#27 degraded by 0.4% in 5.6 years ($-0.07\%year^{-1}$). This is a successful example in long-term instrumental stability. Should the instrument degradation be faster, the statistical treatment would need to be adjusted accordingly. This can be produced by changes in filter transmission, etc. However we must highlight that the instruments used for our analysis exhibited minimum degradation, thus instrumental issues can be discarded to distort the statistics presented for the stations.





Once the slow temporal trend is taken into account, we can try to quantify the residuals in $V_0$ determination, as a quantification of the accuracy of the Langley calibration at the site. The histogram of the $V_0$ values from the PFR (500nm wavelength), normalized to the long-term temporal trend, is provided in Figure 5a (morning Langleys only). The average of the $V_0$ distribution is 1.0 and the standard deviation is $\sigma = 0.0033$. In the plot we have superimposed a Gaussian distribution with the same

mean and standard deviation. The $V_0$ distribution does not pass a normality test mainly because the distribution has strong kurtosis (leptokurtic shape), with up to 81% of the data contained in $\pm 1\sigma$, indicating that most of the values are very close to the average. The standard deviation (0.3%) is therefore a reasonable (even conservative) estimation of the calibration uncertainty at Mauna Loa, that agrees with the uncertainty reported by Holben et al. (1998) for AERONET. The same analysis for Izaña was carried out with the data of Cimel #244, that is operated continuously since November 2011. The histogram of the residuals

of the linear fit of $V_0$ is depicted in Figure 5b, with a relative standard deviation of 0.0046 (or 0.5%). The distribution of the residuals at Izana follows a Gaussian distribution (at 95% confidence level). This particular instrument (in the 500nm channel) degraded by 0.35% in 5 years ($-0.07\% year^{-1}$), thus showing also high stability.

Furthermore, we can evaluate the statistical uncertainty of the $V_0$ determination as a function of the number of averaged Langley plots, with respect to the linear interpolation described before. For this purpose, we have computed moving averages

between 5 and 30 days (number of Langley $V_0$'s), and compared them with the reference value obtained from the linear interpolation. The 15-day moving average is also plotted in Figure 4. We basically calculate the residuals between the moving averages and the linear temporal trend, as a function of the number of Langley plots that are averaged. The result can be interpreted as the additional uncertainty that is added to the calibration when we average a limited number of Langley-retrieved $V_0$'s, as compared to the temporal linear fit over a long period (>1 year). Figure 6 shows the decrease in this additional uncertainty as

the number of averaged $V_0$'s increases. Note that using only one Langley plot will typically increase the calibration uncertainty by 0.5% ( 1% in total) even though the linear regression fulfills strict quality criteria. If we average more than 20 Langley plots, then we reduce this additional uncertainty to <0.1%.

We have also tried to quantify the differences that can be found between morning ('am') and afternoon ('pm') Langley plots in terms of accuracy. The criteria applied to select afternoon Langley plots are exactly the same as above, but the number of

25 suitable data gets reduced to 134 days per year at Mauna Loa (a factor 1.8 less). The standard deviation of the $V_0$'s gets also higher for 'pm' Langleys ($\sigma = 0.0045$). At Izaña the decrease of 'pm' successful Langleys is not that large, with 155 days per year (a factor 1.2 less), and the standard deviation of the $V_0$'s increases up to 0.006.

The strong requirement in AOD is needed to achieve the high accuracy required by AERONET and GAW-PFR. A recent work by Barreto et al. (2014) included moderate, but stable throughout the day, AOD up to 0.3 in the Langley plot calibrations,

that were used to recover a long-term aerosol optical depth data set at Izaña (spanning 1976-2012) from an astronomical spectrometer. The AOD uncertainty in that case gets increased but it is worth mentioning that, depending on the instrument or the intended application, the set of criteria (for instance in AOD) used to select Langley calibrations can be changed.

Finally it must be noted that the uncertainty estimations have been done for the 500nm wavelength. The standard deviation of the $V_0$'s in a typical $\sim 20 - 30$ Langley series is larger in the UV, at $\sim 0.4 - 0.5\%$, and smaller in the NIR wavelengths (870,

1020, 1640 nm) at $\sim 0.1 - 0.2\%$. This wavelength dependence occurs due to lower AOD at the longer wavelengths. For the UV





the higher variance might be also due to filter blocking issues and also possibly to temperature effects for AERONET Cimels that have not been accounted for in the UV wavelengths (in addition to higher AOD in the UV range) .

### 4.3   Additional uncertainty sources

In order to make a deep assessment of the calibration accuracy using the Langley plot method, we have investigated other

5   possible sources contributing to the uncertainty. First, we have analyzed the variability of the solar extraterrestrial irradiance, which is assumed as constant in our previous analysis. The measurements of the space-based photometer run by PMOD/WRC as part of the VIRGO Experiment on the ESA/NASA SOHO Mission (Fröhlich et al., 1995) were used for this purpose. The VIRGO data series comprises more than 20 years of total and spectral (in three bands) solar irradiance. It clearly shows the 11-year cycle in solar irradiance, which is in the order of 0.1%. Given the frequency of recalibration (at least 2-3 times per

10   year) of the GAW-PFR and AERONET reference instruments, this solar cycle should not be an issue for AOD calculations.

However short-term variations in spectral solar irradiance can be as large as 0.5% (at 402 nm) in few weeks during high solar activity, as it is the case of the episode in October-November 2003, depicted in Fig. 7 for the three Sunphotometer wavelengths (402, 500 and 862 nm). We analyzed the extraterrestrial signal provided by the PFR and the Cimel from the ground during this event, unsing the Langley plot method. The resulting (normalized) $V_0$'s, also included in Fig. 7, are however rather noisy and

do not correlate with the space-based data. Either the atmospheric variability or the instrument precission prevent the detection of this kind of abrupt changes in solar irradiance even from high altitude stations, at least with these particular instruments. Averaging Langley calibration over several weeks is shown again necessary to overcome this possible uncertainty.

Another source of uncertainty that has been analyzed is the presence of the subtropical jet above Izaña in spring, which introduces strong turbulence around 12 km height. This phenomenon is well known by the astronomers of the nearby Canary

Astrophysics Institute, since it produces blurring and twinkling of stars due to turbulent mixing in the Earth's atmosphere, that causes variations of the refractive index. To investigate this, we have analyzed the $V_0$ repeatability as in Fig. 5 but making monthly statistics, in order to check for any seasonality in the quality of the calibrations. The result is shown in Fig. 8. The variability of the Langley plots, as evaluated from the standard deviation of the $V_0$'s (500nm wavelength), is somewhat larger in March and October-November, as compared to the rest of the year. According to Fig. 3 in (Rodríguez-Franco and Cuevas,

2013), March-April are the months with stronger winds in the upper troposphere above Izaña station, but the $V_0$ variability is not conclusive to confirm or discard the hypothesis. This assessment will need further investigations, for instance using other instruments with very high sensitivity like the Brewer spectrophotometer, which is routinely operated at Izaña and calibrated for AOD with the Langley plot method (Lopez-Solano, 2017). But at least we can conclude that noisier Langley plots are to be expected at Izaña in March and fall. At Mauna Loa the standard deviation of the Langley plots is only higher in April, in

coincidence with the higher mean AOD in this month.



## 5 Summary and conclusions

The main aerosol optical depth characteristics of the high elevation sites Mauna Loa and Izaña have been analyzed in detail, in order to quantify the characteristics of these locations for Langley plot calibration of Sun photometers. For this purpose, we used long-term records of AERONET and GAW-PFR reference Sun photometers.

The aerosol monthly climatology derived from both networks agrees within 0.0035 optical depth and shows very low aerosol concentrations. For background conditions used in Langley calibrations, AOD (500nm) ranges from 0.01 to 0.02 for both stations. The seasonality is characterized by a spring maximum at Mauna Loa and the occurrence of Saharan dust events in summer at Izaña. Despite the different network operation (frequent swap of AERONET masters, long deployments for GAW-PFR), and data processing schemes (including cloud-screening), they are both shown to be successful in accurate aerosol
monitoring in such pristine locations.

The analysis of cloudiness was accomplished by means of the cloud products OCRA and ROCCIN from GOME-2. On average, Mauna Loa and Izaña exhibit 24 and 20 clear sky days per month, respectively (very thin cirrus clouds are not included in these statistics). Therefore the clouds are not an obstacle for acquiring sufficient number of Langley plots. If we look for days fulfilling also the requirement of high atmospheric stability needed for accurate Langley plot calibration, we yield to a
climatological average of 243 morning and 134 afternoon periods per year at Mauna Loa (about 20 morning and 11 afternoon Langleys per month). Izaña meets the criteria in 187 morning and 155 afternoon periods (about 15 morning and 13 afternoon Langleys per month on average). These conditions were established for Langley plots having standard deviation of the residuals $\sigma < 0.006$, number of valid points $> 33\%$ of the initial number of direct Sun observations and $AOD(500nm) < 0.025$. Dust events at Izaña (especially in summer), reduce the number of available calibration days during those months but do not prevent
from having a sufficient number of clean days for Langley calibration (13 morning Langleys in August at the minimum).

Despite the strict criteria used to select individual Langley plots, a certain noise derived from small changes in Atmospheric transmission, results in the time series of extraterrestrial signals to have a certain variability. This dispersion has been used to statistically provide a quantification of the calibration accuracy, conservatively estimated as ~0.3% at Mauna Loa and ~0.5% at Izaña for 500 nm, regardless of the data set (GAW-PFR or AERONET). The necessary averaging of Langley-derived extrater-
restrial signals may be replaced by a temporal linear fit in case of long deployments. With these criteria, a single Langley plot will be typically within 1% of the mean.

Due to convective activity, morning Langley plots more often fulfill the prescribed stability conditions than afternoons. The probability to have changes in atmospheric transmission is larger in the afternoons and therefore the noise in extraterrestrial signal determination is also larger as compared to the mornings. This effect has been quantified in terms of reduction in the
number of available accurate Langley plots: at Mauna Loa, a factor 1.8 less afternoon Langleys; and smaller reduction (a factor 1.2 less) is found at Izaña. It has also been found that fast variations in solar extraterrestrial irradiance, up to 0.5% in few weeks, are not easily detectable from the ground with this kind of instruments. Furthermore, the subtropical jet above Izaña is pointed out as possible explanation for the increase in the Langley plot residuals in this station during the spring months.



With this analysis we can conclude that the high-altitude stations Mauna Loa and Izaña meet the GAW-PFR and AERONET network requirements in terms of accuracy , i.e. 0.2-0.5% in calibration factors or 0.002-0.005 in AOD (for $airmass = 1$). The long-term operation and maintenance of reference instruments at these unique locations is shown to be key in accurate aerosol monitoring worldwide. The stability of the reference instruments has also been proved to be very high, with signal loses due to

5   degradation of optical components below 0.1% per year over long periods.

*Acknowledgements.*   The authors gratefully acknowledge the effort of NOAA and AEMET to maintain the Mauna Loa and Izaña observatories. We thank the two site operators for their efforts on day to day instrument maintenance over years. Dr. Antón and Dr. Loyola provided the satellite data for the cloud analysis. We thank Dr. V. Freudenthaler for his advice on the error analysis. This research has received funding from the European Union's Horizon 2020 Research and Innovation Programme under grant agreement No 654109 (ACTRIS-2). The funding

10  by MINECO (CTM2015-66742-R) and Junta de Castilla y León (VA100P17) is also acknowledged.



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



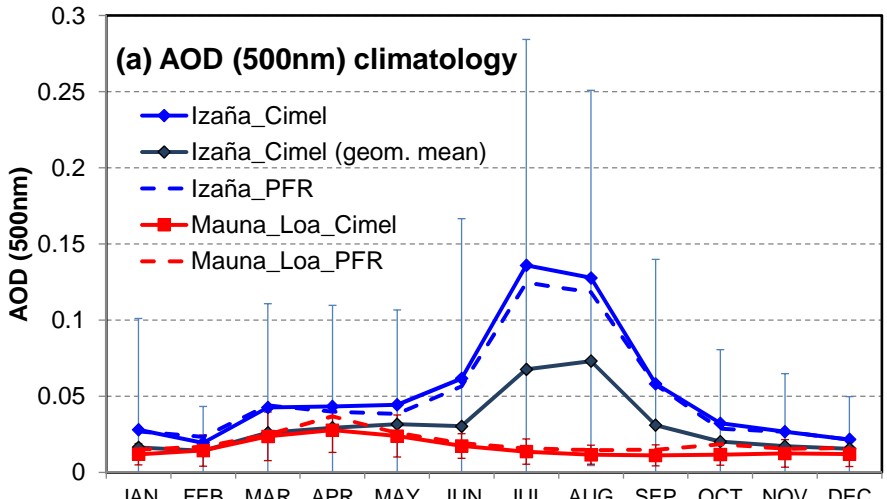

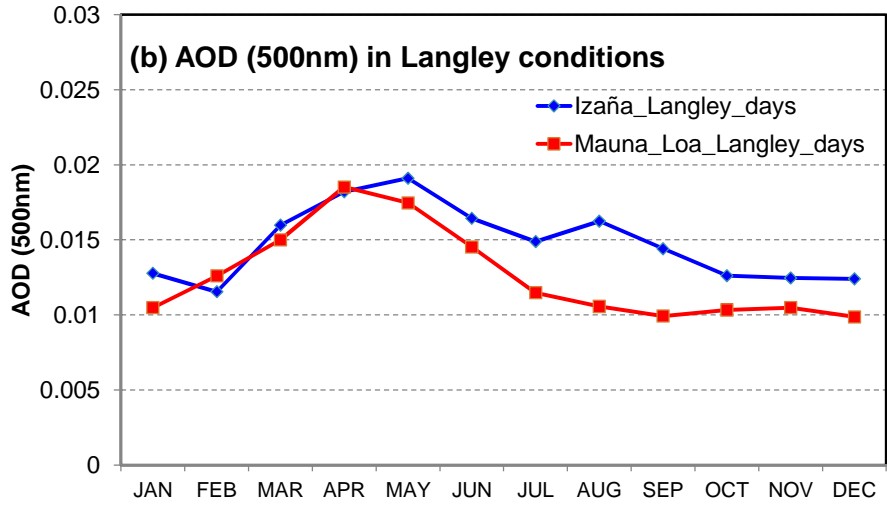

**Figure 1.** (a) Monthly mean aerosol optical depth (500nm) at Mauna Loa (1994-2016) and Izaña (2004-2016) for AERONET and GAW-PFR. Bars indicate ±1 monthly standard deviation. Black line indicates geometric mean values for AOD at Izaña (in contrast to the arithmetic mean for the other variables). (b) Monthly mean aerosol optical depth (500nm) for the days fufilling the criteria for Langley calibration as given in section 4.1.



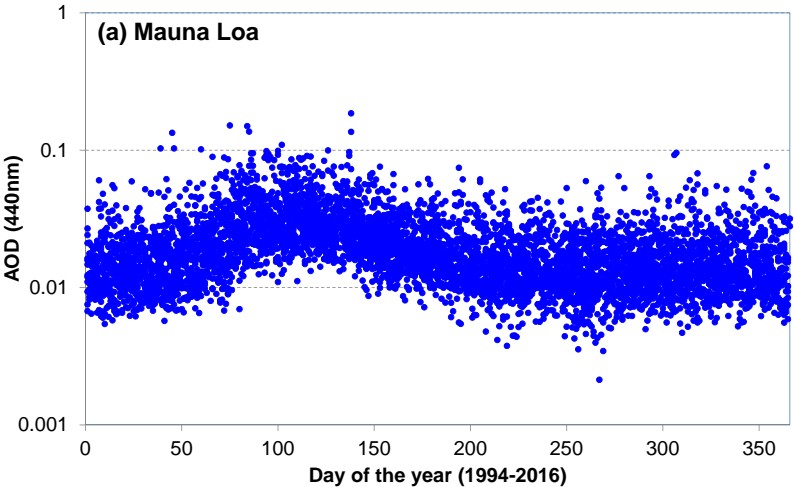

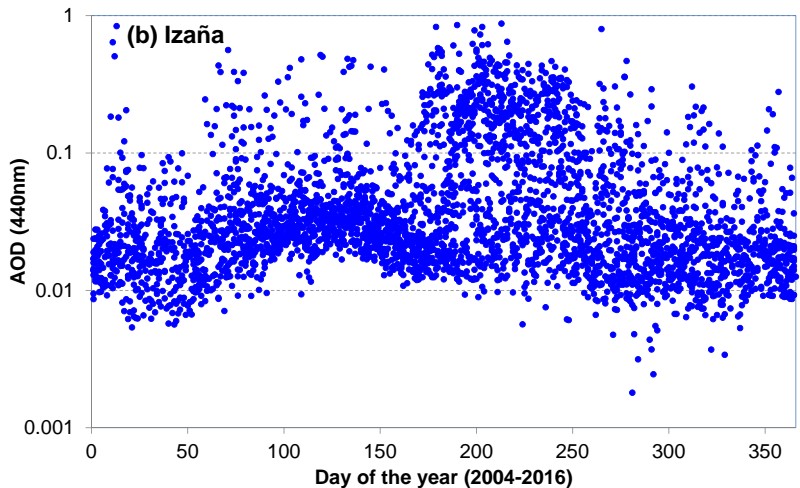

**Figure 2.** Daily means of aerosol optical depth (440nm) as a function of the day of the year at: (a) Mauna Loa (1994-2016) and (b) Izaña (2004-2016) using AERONET data.




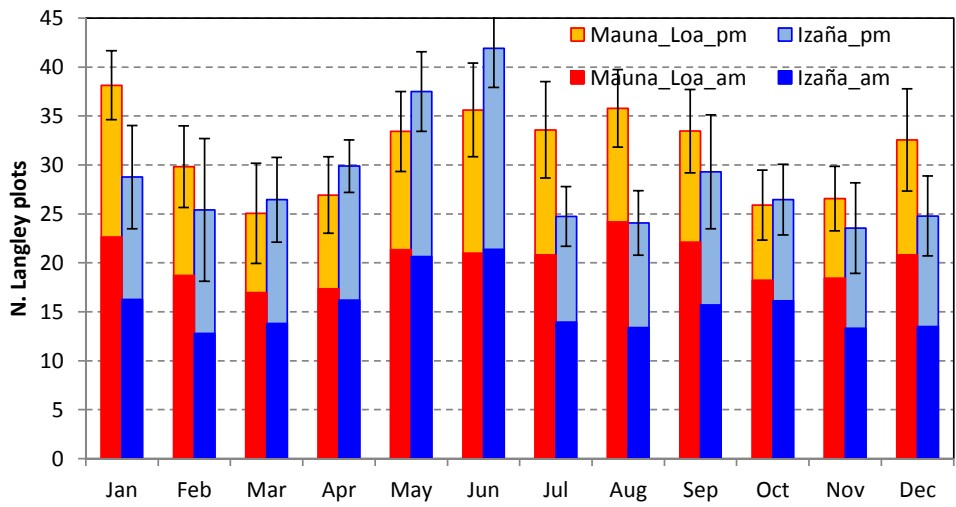

**Figure 3.** Mean number of suitable Langley calibrations per month at Mauna Loa and Izaña based on GAW-PFR and AERONET data (see text). Bars indicate ±1 standard deviation within the month. Morning ('am') and afternoon ('pm') Langley plot calibrations are given separately.



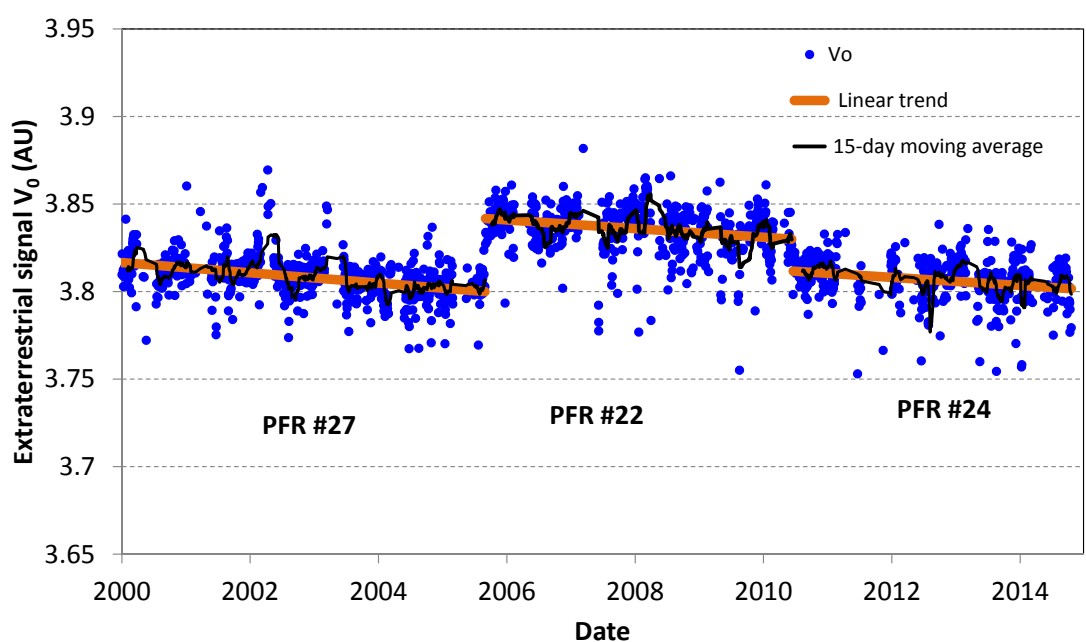

**Figure 4.** Daily extraterrestrial voltages ($V_0$) at 500nm wavelength obtained with the Langley plot method for the GAW-PFR at Mauna Loa (morning calibrations only). The temporal linear fit to the $V_0$'s for each instrument deployment is superimposed, as well as the 15-day moving average. Note that these are instrument signals, i.e. depend on each particular instrument and are not directly comparable.




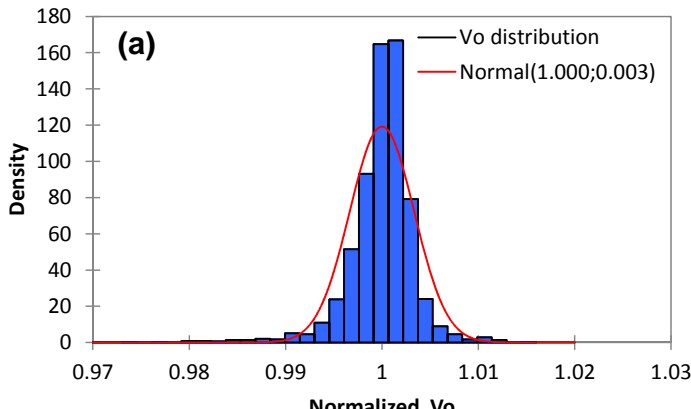

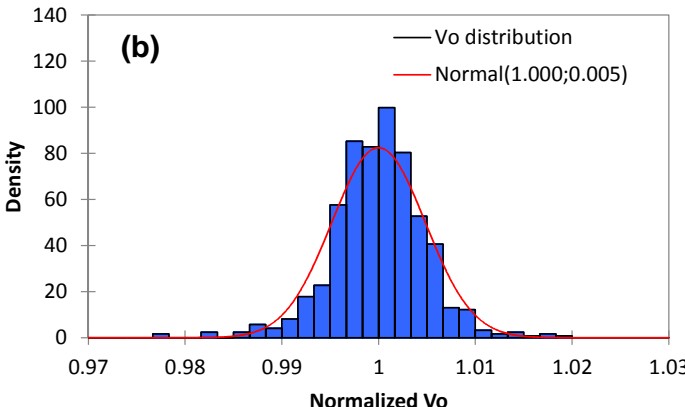

**Figure 5.** Histogram of daily extraterrestrial voltages ($V_0$) at 500nm wavelength normalized by the temporal trend: (a) At Mauna Loa using GAW-PFR data (2000-2014); (b) At Izaña using AERONET #244 (2012-2016). Red lines indicate a normal distribution (with the given parameters).



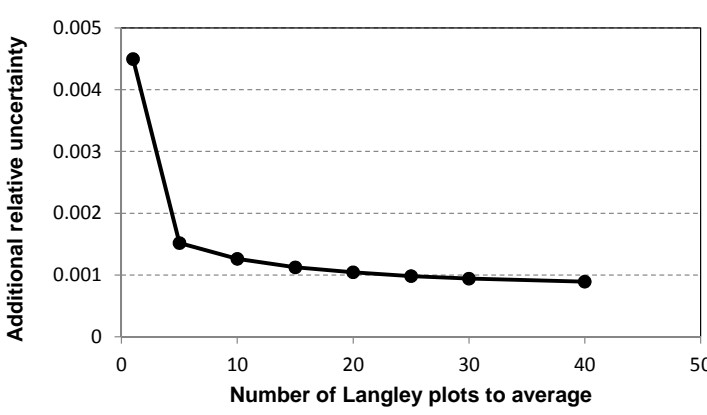

**Figure 6.** Additional uncertainty added to the Langley plot calibration vs. number of Langley plot $V_0$'s that are averaged, using GAW-PFR data (500nm) at Mauna Loa (2000-2014).

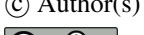


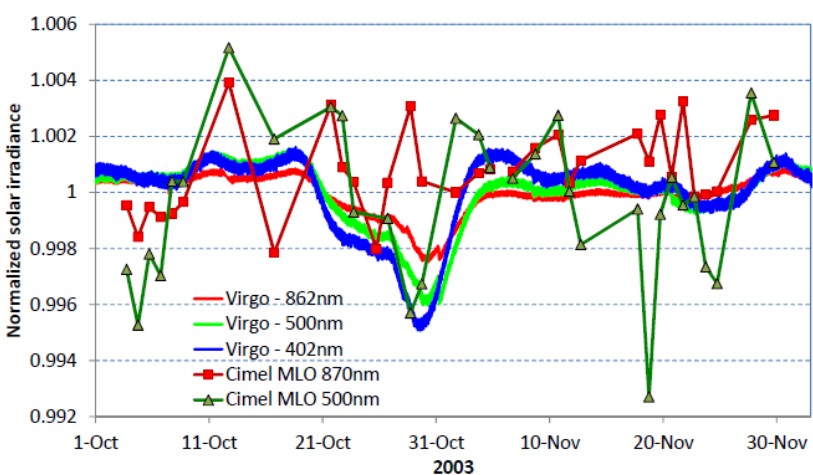

**Figure 7.** Solar extraterrestrial normalized irradiance as measured by the VIRGO space-based photometer during 2003-2004 at three wavelengths: 402nm (blue), 500nm (green) and 862nm (red).



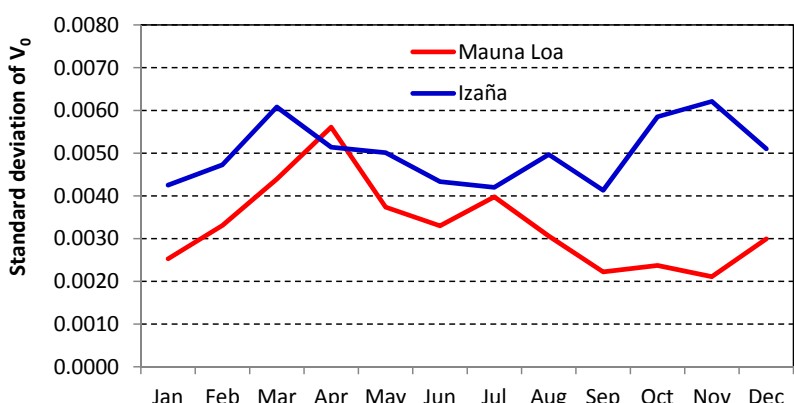

**Figure 8.** Standard deviation of $V_0$'s (500nm wavelength) from Langley calibrations for every month at Izaña and Mauna Loa using AERONET data.



**Table 1.** Cloud fraction and number of clear sky days over Mauna Loa and Izaña observatories, derived from GOME-2 cloud products (Loyola R. et al., 2010) over 2007-2014 . Clear sky is defined as cloud fraction $< 0.1$. The number of investigated days within each month for the 8-year period is also provided.

| | Mauna Loa | | | | Izaña | | | |
|---|---|---|---|---|---|---|---|---|
| | Mean Cloud cover fraction | Frequency of cloud cover $< 0.1$ (%) | Mean N. days fraction$<$ 0.1 | N days | Mean Cloud cover fraction | Frequency of cloud cover $< 0.1$ (%) | Mean N. days fraction$<$ 0.1 | N days |
| Jan | 0.06 | 88.9 | 28 | 162 | 0.13 | 60.67 | 19 | 178 |
| Feb | 0.11 | 75.0 | 21 | 164 | 0.12 | 67.96 | 19 | 181 |
| Mar | 0.14 | 70.8 | 22 | 171 | 0.14 | 61.22 | 19 | 196 |
| Apr | 0.11 | 76.1 | 23 | 155 | 0.11 | 58.48 | 18 | 171 |
| May | 0.06 | 81.9 | 25 | 171 | 0.08 | 68.85 | 21 | 183 |
| Jun | 0.05 | 85.6 | 26 | 160 | 0.05 | 80.56 | 24 | 180 |
| Jul | 0.03 | 86.2 | 27 | 159 | 0.06 | 76.24 | 24 | 181 |
| Aug | 0.02 | 91.2 | 28 | 159 | 0.08 | 66.47 | 21 | 173 |
| Sep | 0.07 | 79.2 | 24 | 149 | 0.14 | 53.29 | 16 | 167 |
| Oct | 0.09 | 76.9 | 24 | 156 | 0.16 | 58.48 | 18 | 171 |
| Nov | 0.12 | 72.9 | 22 | 155 | 0.15 | 57.74 | 17 | 168 |
| Dec | 0.19 | 68.2 | 21 | 157 | 0.16 | 60.34 | 19 | 174 |
| YEAR | 0.09 | 79.4 | 290 | 1918 | 0.11 | 64.34 | 235 | 2123 |



**Table 2.** Deployment periods of GAW-PFR instruments at Mauna Loa and Izaña.

| (a) Mauna Loa | | | |
|---|---|---|---|
| Instrument | Start date | End date | N days |
| PFR #27 | 1-Jan-2000 | 1-Sep-2005 | 2070 |
| PFR #22 | 2-Sep-2005 | 16-Jun-2010 | 1748 |
| PFR #24 | 16-Jun-2010 | 31-Dec-2014 | 1659 |
| (b) Izaña | | | |
| Instrument | Start date | End date | N days |
| PFR #25 | 9-Jun-2001 | 25-Feb-2009 | 2818 |
| PFR #06 | 14-May-2009 | 1-Jan-2013 | 1328 |
| PFR #21 | 2-Jan-2013 | 30-Apr-2014 | 483 |
| PFR #06 | 1-May-2014 | 31-Dec-2014 | 244 |