# Peer review of "Assessment of Sun photometer Langley calibration at the high-elevation sites Mauna Loa and Izaña"

_Atmospheric Chemistry and Physics, 2018_

## Referee Comment (RC1) · Anonymous Referee #1 · 8 Jun 2018

The paper is a very useful document describing and characterizing the most important calibration sites for photometers, Izana and Mauna Loa. It is really well written, simple and schematic, and provides all the necessary information for scientists involved in photometer science. Therefore I consider it suitable for the publication.

Below few minor revisions to be done: 1. page 4, lines 17-19: it is stated that "Possible instrument fluctuations due to shipping are controlled by using always a couple of masters that travel together and rigorous comparison of master instruments at the inter-calibration sites". This is a good solution for the stability of the master instruments. However the other equipments shipped with a round trip for calibrating against

the master can suffer of the same problem, and come back operative with a calibration no more perfect as the one done in the calibration site. Has this point never been checked or studied ?

2. page 5 lines 12-15, add that the CAELIS software will be better described in section 4.1

3.Page 8 line 5: in the text it is stated that the error bar is the year-to-year variability, but in the caption of Figure 3 it is stated that the monthly mean is. Please clarify.

4. section 4.2, Figure 4 : It would be interesting to highlight if the yearly variability has a sort of seasonal dependence. In this case it could depend on internal temperature, not well kept constant, or in the parametrization assumed in the Lambert-Beer Law. Which correction ar assumed for gases absorption? it would be interesting they are described briefly in the text.

5. Capition Figure 1: fuLfilled

---

## Referee Comment (RC2) · Anonymous Referee #2 · 12 Jun 2018

Review of Toledano et al., "Assessment of Sun photometer Langley calibration at the high-elevation sites Mauna Loa and Izana"

**General comments**

In general, analyzing long and high quality time series of different instruments operated independently in two different networks is valuable work for the atmospheric physics community and appropriate for publication in ACP.

The manuscript describes the environmental variables at Mauna Loa and Izana, compiles climatologies of aerosol optical depth and finally aims at an analysis of the Langley calibration uncertainty for these two sites.

So the manuscript covers a wide scope, however, the scientific impact is weakened by a lack of a rigorous, in-depth analysis. In particular, the statistical (uncertainty) analysis includes several issues. A general indicator for this weakness is that uncertainty and accuracy are often used synonymously, systematic and statistical errors are not treated separately. Without any additional and more detailed discussion, accuracy should be replaced by uncertainty throughout the manuscript.

**Specific comments**

Section 2.1

P3, L4. "…because the AOD is very low and stable". In fact, I believe, the AOD variability is the actual criterion rather than just low AOD (although typically, both are correlated). This slight misconception appears again later in the manuscript.

P4, L3. Just as a question, I wonder why the data in Table 1 are not displayed as e.g. a bar chart? This would probably even save space and convey the information much easier. Then again, I would argue that cloudy periods are mutually exclusive from AOD measurement periods, so cloud statistics do not add any information for the conclusions here, if later, statistics on Langley days are shown anyways. One interesting insight from cloud information could be the probability of suitable Langley conditions in cloud free conditions.

Section 2.2

P5, L10. Table 2. Again, just a suggestion, but I believe that visual timelines of the instrument deployment (e.g. in the style of a Gantt chart) would be a lot more efficient than just printing numbers in Table 2. Note that "table" should be capitalized when followed by a number.

Section 3

P5, L27pp. If a quantitative comparison of time series is the goal here then the same period should be compared (rather than 1994-2016 versus 2000-2014), otherwise the discussion about other causes for differences is problematic. Regarding the cloud screening methods, as far I am aware of, at least part of the cloud screen for the PFR is based on Smirnov's method. What systematic difference

can be expected from the differences of the methods, i.e. is Wehrli's method more stringent and therefore filters more data points (possibly with a bias of higher AOD)?

P6, L21. In fact, it looks more like a bimodal distribution, rather than log-normal.

P7, L3. Why is the AOD in Fig. 2 not shown for 500 nm, as in Fig. 1? Also, what is the reason of using a log scale here and not in Fig. 1? In addition, I would like to suggest that for Fig. 2 histograms would be better suited to reveal the distributions.

Section 4.1

P7, L28. Surely the criteria also affect the number of suitable Langley plots and hence are relevant in the "climatological sense".

Section 4.2

This complete section should be improved by reducing confusing and irrelevant sentences and sharpening the statistical argumentation.

P8, L17. Of course there is no physical measurement without uncertainty. Is the "noise caused by changes in atmospheric transmittance having a hyperbolic (…) dependence" mainly due to residual AOD variations, which affect the slope and/or y-intersect of the Langley plots?

P8, L30. This a confusing paragraph. The sentence "Should the instrument degradation…" can be safely omitted. What is the significance of the sentence "…instrumental issues can be discarded…"? In fact, the linear trend is small (but detectable) and has been correctly taken into account.

P9, L3, Fig. 5. For the y-label, change "density" to "N" and also a heading would help like in Fig. 2, indicating site and instrument. Also, why has the analysis been done for Mauna Loa for 14 years and 3 instruments, while for Izana, only for 4 years and 1 instrument?

P9, L14. The concept of "adding statistical uncertainty" is statistically confusing and the representation in Fig. 6 is suboptimal in many ways.

First, to avoid this confusion, I believe, simply the absolute uncertainty should be considered and plotted here. Also, how does the uncertainty of a one day Langley plot (as shown in Fig. 5) increase to "1% in total". Please clarify.

Second, the statistical uncertainty is generally expected to decrease with square root N, the number measurements, in this case number of days. So the data would ideally be plotted in log-log scale to be able to compare it to a linear slope of -0.5. A deviation from that slope indicates additional error sources (short term drift of the instrument or changes in the signal).

Third, the region between 1 and 10 days seems important, so more data points would be beneficial.

P9, L24. Please explain why suitable days get reduced.

P9,L28. Please clarify the "strong requirement" and include the variability of the AOD, rather than just low AOD.

P9, L35. Again, it is the lower variability of AOD and the wavelength dependence is caused by the Angstrom exponent >0.

Section 4.3

For a "deep assessment" a lot more factors should at least be mentioned. E.g. gas absorption of ozone at 500nm, how is the ozone considered, climatological values?

Or, e.g. what is the effect of different definitions of air mass? As mentioned, it becomes important for large air masses.

P10, L11, Fig. 7. 401 nm is not relevant here, so it should be omitted for clarity of the figure.

Considering the standard deviation for MLO in Fig. 5 of 0.3% it is not surprising that variations at the 0.4 % level are not significant and that there are "no correlations". Plotting error bars or bands for the Cimel AOD in Fig. 7, may visually reduce the expectation to detect correlations.

Also, why not use a 2-day moving average? Maybe the dip around the 31.10. would actually be significant in the 500 nm Cimel calibrations.

P10, L17. "…averaging over several weeks". From Fig. 6, it looks like averaging more than 10 days does not significantly reduce the uncertainty.

P10, L18. Could the authors please explain the physical reason why turbulence at 12 km altitude and variations in the refractive index should have an effect on the AOD? Surely this would affect the imaging of stars, but does the blurring effect cause direct solar radiation to be scattered outside the FOV of the sun photometers?

P10, L27. Would a Brewer really be better suited for this study? Is the sensitivity of the instrument an issue here, or the stability?

Section 5

P11, L23. At least in the conclusion, the statistical uncertainty should be clearly specified as 1-sigma standard deviation for a one day Langley plot. From Fig. 5, this was estimated to 0.3% (Mauna Loa) and 0.5% (Izana) . "…a single Langley plot will be typically within 1% of the mean". What exactly does the 1% signify? 95% confidence interval? Why should the averaging be replaced by the temporal linear fit? Fitting a straight line would be a generalized method, including averaging as a special case with a line with slope zero.

P11, L32. The discussion about the subtropical jet was not really conclusive.

**Technical corrections**

There are different rules about capitalization, but I think in the context of e.g. "direct sun measurements", "sun photometry" etc., the common practice is to not capitalize "sun".

P1,L11. "…this uncertainty being smaller…"

P3, L15. "…it is the reference observatory…"

P4, L18. "…direct sun signal…"

P6, L2. "Smirnov"

P8, L9. "a certain", or even better "a slight"?

P11, L14. "…we find a climatological average…"

P12, L4. "signal losses"

---

## Author Comment (AC1) · 20 Aug 2018

The paper is a very useful document describing and characterizing the most important calibration sites for photometers, Izana and Mauna Loa. It is really well written, simple and schematic, and provides all the necessary information for scientists involved in photometer science. Therefore I consider it suitable for the publication.

Below few minor revisions to be done:
1. page 4, lines 17-19: it is stated that "Possible instrument fluctuations due to shipping are controlled by using always a couple of masters that travel together and rigorous comparison of master instruments at the inter-calibration sites". This is a good solution for the stability of the master instruments. However the other equipments shipped with a round trip for calibrating against the master can suffer of the same problem, and come back operative with a calibration no more perfect as the one done in the calibration site. Has this point never been checked or studied ?
For masters (which are well controlled) it is very rare that they change calibration during transport. For field instruments we carefully check the data on site after deployment in the field. Any anomalous behavior like AOD dependence on airmass, negative AOD, anomalous fluctuations in the triplets, etc. would be indicative of some problem during transport or installation. A set of flags is operational in AERONET to detect all kind of (known) instrument malfunctions from the data.

2. page 5 lines 12-15, add that the CAELIS software will be better described in section 4.1
Added.

3. Page 8 line 5: in the text it is stated that the error bar is the year-to-year variability,but in the caption of Figure 3 it is stated that the monthly mean is. Please clarify.
In this figure we show the number of suitable Langley calibrations in a certain month over a multiannual period. For a given month, e.g. January, each year has a different number of Langley days. Therefore we provide the average and the standard deviation that arise from the year to year variability. A clarification has been added in the caption.

4. section 4.2, Figure 4 : It would be interesting to highlight if the yearly variability has a sort of seasonal dependence. In this case it could depend on internal temperature, not well kept constant, or in the parametrization assumed in the Lambert-Beer Law. Which correction ar assumed for gases absorption? it would be interesting they are described briefly in the text.
It's not very likely that residual temperature dependence could result in the slight seasonal dependence of figure 4, because the internal temperature of the PFR is continuously monitored and the optical elements (photodiode and interference filters) are temperature controlled to better than 0.1 K. The expected small seasonal changes in atmospheric gases are not likely to bias the Langley plots either, because Langleys are not affected by the amount of the absorber but would be by a systematic diurnal cycle.
No gases absorption correction is assumed in the Langley analysis to obtain the extraterrestrial signals (only total optical depth is derived). For the aerosol optical depth climatology in section 3 (figs. 1 and 2) we used the standard corrections of GAW-PFR and AERONET networks (see references in the manuscript). AERONET basically uses climatology tables for ozone and NO2, whereas ozone from OMI is used in GAW-PFR. This information has been added to the text.

5. Caption Figure 1: fulfilled
Corrected.

Anonymous Referee #2

Review of Toledano et al., "Assessment of Sun photometer Langley calibration at the high-elevation sites Mauna Loa and Izana"

**General comments**
In general, analyzing long and high quality time series of different instruments operated independently in two different networks is valuable work for the atmospheric physics community and appropriate for publication in ACP.
The manuscript describes the environmental variables at Mauna Loa and Izana, compiles climatologies of aerosol optical depth and finally aims at an analysis of the Langley calibration uncertainty for these two sites.
So the manuscript covers a wide scope, however, the scientific impact is weakened by a lack of a rigorous, in-depth analysis. In particular, the statistical (uncertainty) analysis includes several issues. A general indicator for this weakness is that uncertainty and accuracy are often used synonymously, systematic and statistical errors are not treated separately. Without any additional and more detailed discussion, accuracy should be replaced by uncertainty throughout the manuscript.

**Specific comments**
Section 2.1
P3, L4. "…because the AOD is very low and stable". In fact, I believe, the AOD variability is the actual criterion rather than just low AOD (although typically, both are correlated). This slight misconception appears again later in the manuscript.
Yes, the misconception has been removed. The very low AOD makes it possible that it is very stable in absolute sense, but the crucial point for a Langley plot is the stability. We state this at the very beginning of the introduction (P1, L16).

P4, L3. Just as a question, I wonder why the data in Table 1 are not displayed as e.g. a bar chart? This would probably even save space and convey the information much easier. Then again, I would argue that cloudy periods are mutually exclusive from AOD measurement periods, so cloud statistics do not add any information for the conclusions here, if later, statistics on Langley days are shown anyways. One interesting insight from cloud information could be the probability of suitable Langley conditions in cloud free conditions.
The low cloudiness is an important characteristic of the observatories for solar radiation observations. Previously to the Langley day analysis, we try to show the main features of the station locations for them to be so unique, including this cloud statistics and the aerosol optical depth climatology. The current table contains 4 pieces of information per station, a bar chart would probably be too busy.
The probability of suitable Langley conditions in cloud free conditions, as derived from the database, is about 83% at Mauna Loa and 79% at Izaña. This is seasonal dependent for Izaña, where the probability is about 60% in July and August due to Saharan dust.

Section 2.2
P5, L10. Table 2. Again, just a suggestion, but I believe that visual timelines of the instrument deployment (e.g. in the style of a Gantt chart) would be a lot more efficient than just printing numbers in Table 2. Note that "table" should be capitalized when followed by a number.
"Table" has been capitalized as commented. A figure would be more visual for the timelines but the exact information about number of deployment days would be lost. Moreover there is some visualization in Figure 4, so we would prefer to leave it as is now.

Section 3

P5, L27pp. If a quantitative comparison of time series is the goal here then the same period should be compared (rather than 1994-2016 versus 2000-2014), otherwise the discussion about other causes for differences is problematic. Regarding the cloud screening methods, as far I am aware of, at least part of the cloud screen for the PFR is based on Smirnov's method. What systematic difference can be expected from the differences of the methods, i.e. is Wehrli's method more stringent and therefore filters more data points (possibly with a bias of higher AOD)?

We try to show that long-term AOD climatologies as derived by GAW-PFR and AERONET are equivalent, despite the different periods and cloud-screening. Both networks accumulate enough observations and come to nearly identical climatology, that's the message we intend, rather than a detailed instrumental / quantitative comparison. Such approach has been accomplished by other publications, some of which are cited for reference.

As explained in Wehrli (2008) one step of the three-step method in GAW-PFR is a temporal filter inspired in Smirnov method, although the different sampling method and thresholds make the results to be probably different. To what extent the cloud-screening methods differ would need a dedicated analysis that is out of the scope of this manuscript. Some comparison and scoring can be found in Kazadzis et al. (2018a).

The reviewer is right that a quantitative comparison in the sense of the WMO threshold of 0.005 + 0.010/airmass does not apply for a climatology (no airmass can be easily attributed to a monthly mean). Therefore we have removed the sentence.

P6, L21. In fact, it looks more like a bimodal distribution, rather than log-normal.

The AOD histograms for Mauna Los and Izaña are closer to log-normal than normal distribution, as explained in the paper by O'Neill. It's true that a bi-modal distribution would also produce the geometric and arithmetic means to be separated. To avoid that possible confusion, the sentence has been changed to:

The geometric mean is often more suitable for AOD statistics, because the lognormal probability distribution is a better reference than a normal distribution for most aerosol types (O'Neill et al., 2000).

P7, L3. Why is the AOD in Fig. 2 not shown for 500 nm, as in Fig. 1? Also, what is the reason of using a log scale here and not in Fig. 1? In addition, I would like to suggest that for Fig. 2 histograms would be better suited to reveal the distributions.

Fig. 2 is based on AERONET data and it is shown to highlight in terms of daily means (each point is a day) the occurrence of enhanced background at both locations in spring and episodic dust at Izaña. The plot is nearly the same with 500nm but doesn't need to be directly comparable with Fig. 1. Wavelength channels are highly correlated anyway. Log-scale in Fig. 2 is used to facilitate the visualization of the different daily mean values.

We have changed the plot to 500nm for the sake of consistency. Histograms do not provide much new information, so they have not been included. They are shown here anyway, and could be included in the supplement if necessary.

[Figure]

Section 4.1

P7, L28. Surely the criteria also affect the number of suitable Langley plots and hence are relevant in the "climatological sense".

Yes, it's true that they affect but we indicate that it's not a critical (significant) change. The impact is not large because Langley days in most cases are selected no matter what (reasonable) criteria are used. But there are always some cases in the borderline that can be in or out depending on the threshold. Obviously different thresholds would yield to different number of Langley plots. This sentence has been added for clarity: "Other thresholds were tested and revealed no significant changes in the analysis."

Section 4.2

This complete section should be improved by reducing confusing and irrelevant sentences and sharpening the statistical argumentation.

P8, L17. Of course there is no physical measurement without uncertainty. Is the "noise caused by changes in atmospheric transmittance having a hyperbolic (…) dependence" mainly due to residual AOD variations, which affect the slope and/or y-intersect of the Langley plots?

Yes, the main reason for changes in transmittance is the AOD variations. A change of 0.005 in AOD (at any wavelength) is clearly possible and that's the reason to use the high-altitude stations: the less aerosol, the lower the absolute magnitude of variation (for a total aerosol optical depth of 0.01, even 50% relative change would only produce a variability of 0.005). Pressure, water vapor, NO2 or ozone variations can also contribute, but the change needed in these components for a significant modification of the extraterrestrial constant (ETC) would be too large except for pressure in the UV channels (see table below). For instance, a 0.5% modification in ETC (at 500nm) would need a change of 35hPa during the morning or afternoon, 150DU of ozone or 1DU of NO2, all of these unrealistic change rates. Similarly, only a change >20mm during the morning or afternoon in the water vapor column would significantly affect 1020 and 1640nm channels. However a change of 5hPa in pressure during the morning or afternoon would be noticeable for ETC in 340 and 380nm.

| | Needed change to produce V0 change of 0.5% | | | | | |
|---|---|---|---|---|---|---|
| Channel | Pressure | Ozone | Water vapor | NO2 | CO2 | CH4 |
| (nm) | (hPa) | (DU) | (cm) | (DU) | (ppm) | (ppb) |
| 340 | 7 | 89 | | 0.5 | | |
| 380 | 11 | 1235 | | 0.3 | | |
| 440 | 21 | 1556 | | 0.4 | | |
| 500 | 35 | 158 | | 1.1 | | |
| 675 | 120 | 122 | | 24.9 | | |
| 870 | 334 | | | | | |
| 1020 | 635 | | 2.1 | | | |
| 1640 | 4387 | | 3.8 | | 486 | 1011 |

Moreover, Marenco (2007) demonstrated that with Langley plots alone it is impossible to identify atmospheric variations having a diurnal periodicity with extreme at noon. This has been added to the text. Some information about the presence of systematic errors has been added to section 4.3 (see corresponding answer below).

P8, L30. This a confusing paragraph. The sentence "Should the instrument degradation…" can be safely omitted. What is the significance of the sentence "…instrumental issues can be discarded…"? In fact, the linear trend is small (but detectable) and has been correctly taken into account.
We have deleted the sentence as suggested.
About the "instrumental issues can be discarded": we wanted to emphasize that the uncertainty in extraterrestrial signals is mainly a result of atmospheric changes, not due to instrumental issues. However, as the reviewer indicates, this possibility has been already explained and the instrumental drift correctly taken into account, therefore we have removed the sentence too.

P9, L3, Fig. 5. For the y-label, change "density" to "N" and also a heading would help like in Fig. 2, indicating site and instrument. Also, why has the analysis been done for Mauna Loa for 14 years and 3 instruments, while for Izana, only for 4 years and 1 instrument?
The changes to the plots have been done.
This kind of analysis is facilitated by long deployment periods, because we try to evaluate drift in instrument extraterrestrial signals. The three long deployments of GAW-PFR data in Mauna Loa (see Fig. 4) are optimal. At Izaña there were multiple changes in GAW-PFR instruments, so we chose the longest deployment available, in this case AERONET #244.

P9, L14. The concept of "adding statistical uncertainty" is statistically confusing and the representation in Fig. 6 is suboptimal in many ways.
First, to avoid this confusion, I believe, simply the absolute uncertainty should be considered and plotted here. Also, how does the uncertainty of a one day Langley plot (as shown in Fig. 5) increase to "1% in total". Please clarify.
This was a mistaken approach, thanks for pointing it out. We assumed the linear trend to have the 0.3-0.5% uncertainty as given by the standard deviation in Fig 5. But the linear trend is almost the "truth" as the error gets divided by Square root N (and for this long term analysis, N is large). We have reformulated the analysis completely.
Fig. 5 actually provides the uncertainty of a single Langley plot calibration. We have now used a 2-sigma criterion (k=2) to provide the uncertainty at the 95% confidence level, i.e. 0.007 or 0.7% for Mauna Loa and 0.9% at Izaña. This is the Type A uncertainty for a single Langley plot. As we combine an increasing number of Langley plots, the standard deviation of the mean gets reduced as in Fig 6, which now shows uncertainty as a function of the number of Langley plots that are averaged, starting with 1 and up to 20. This has been plotted in log-log scale, with the theoretical line of slope= -0.5 plotted as reference. The data fit to slope of -0.40, not far for the expected. We have also included more data in the region between 1 and 10, as indicated.

Second, the statistical uncertainty is generally expected to decrease with square root N, the number measurements, in this case number of days. So the data would ideally be plotted in log-log scale to be able to compare it to a linear slope of -0.5. A deviation from that slope indicates additional error sources (short term drift of the instrument or changes in the signal). Third, the region between 1 and 10 days seems important, so more data points would be beneficial.
(See also answer above). As mentioned, the plot in log-log scale has a slope of -0.40, so it's not far from the expected value. The existence of additional error sources cannot be discarded, as

well as possible correlations. The long-term stability of the instruments is demonstrated but short-term drifts are possible due to small obstructions, residual temperature changes, etc.

P9, L24. Please explain why suitable days get reduced.
We meant that the number of suitable pm langleys is only 134 days per year at Mauna Loa.

P9,L28. Please clarify the "strong requirement" and include the variability of the AOD, rather than just low AOD.
The AOD requirement is AOD(500nm)<0.025. We agree that it is necessary to specify that the threshold in AOD is intended to reduce the possibility of AOD variability, rather than AOD itself. New sentence is: "The strong requirement of AOD(500nm) < 0.025 is needed to prevent AOD variability and achieve the low uncertainty required by AERONET and GAW-PFR."

P9, L35. Again, it is the lower variability of AOD and the wavelength dependence is caused by the Angstrom exponent >0.
The sentence has been changed to: "This wavelength dependence in uncertainty occurs due to lower AOD variability at the longer wavelengths."

Section 4.3
For a "deep assessment" a lot more factors should at least be mentioned. E.g. gas absorption of ozone at 500nm, how is the ozone considered, climatological values?
Or, e.g. what is the effect of different definitions of air mass? As mentioned, it becomes important for large air masses.
We have included a new paragraph in this section, providing the main factors that affect Langley plots, with a set of references that extensively describe the effect of finite bandwidth, contribution of diffuse light, airmass and systematic semidiurnal variations of aerosol. The quantification made e.g. by Reagan (1986) indicates that these errors should not exceed 0.1% with the specified instruments (field of view 1.2deg., GPS time, etc.) and Langley conditions at the high altitude stations.
About other components, there is literature about the semidiurnal pressure variation ("atmospheric tide", Le Blancq, 2011, about 1-2 hPa amplitude). The current processing accounts for pressure changes and they would mean too little change in Rayleigh optical depth anyway. The diurnal changes in water vapor (in the order of few mm amplitude), would not significantly affect the aerosol channels. Diurnal cycles in other components such as NO2, CO2 or ozone, linked to incoming radiation, vegetation activity, etc. are also of small amplitude and therefore are not expected to produce significant bias in Langley plot calibrations.

P10, L11, Fig. 7. 401 nm is not relevant here, so it should be omitted for clarity of the figure.
We have removed it for clarity, although we lose information on the changes in the blue channel. Most of the discussion is focused on the 500nm channel, but both PFR and Cimel have UV channels.

Considering the standard deviation for MLO in Fig. 5 of 0.3% it is not surprising that variations at the 0.4 % level are not significant and that there are "no correlations". Plotting error bars or bands for the Cimel AOD in Fig. 7, may visually reduce the expectation to detect correlations. Also, why not use a 2-day moving average? Maybe the dip around the 31.10. would actually be significant in the 500 nm Cimel calibrations.
The error bars have been added to the Cimel data at 500nm. The moving average does not improve much because the variations in the Cimel calibrations around 5.10 and 18.11 are not at all shown by Virgo data, even if the dip around 31.10 is a bit more clear (only for 500nm, not for 870nm).

P10, L17. "…averaging over several weeks". From Fig. 6, it looks like averaging more than 10 days does not significantly reduce the uncertainty.

The sentence was not very clear. We meant that, in order to accumulate 10-15 good Langleys, it's possible that 2-3 weeks of measurements are needed. The sentence has been rewritten.

P10, L18. Could the authors please explain the physical reason why turbulence at 12 km altitude and variations in the refractive index should have an effect on the AOD? Surely this would affect the imaging of stars, but does the blurring effect cause direct solar radiation to be scattered outside the FOV of the sun photometers?

The aerosol optical depth depends almost linearly with the scattering coefficient, which in turn depends on the refractive index and size distribution. For fine particles, the influence of the refractive index on the AOD can be large (see for instance the simulations in Wang & Rood, JGR2008).

P10, L27. Would a Brewer really be better suited for this study? Is the sensitivity of the instrument an issue here, or the stability?

Both sensitivity and stability are pertinent, as well as sampling speed and frequency, and other instrumental factors, corrections, etc. For the sake of clarity, we have removed the sentence about the Brewer instrument.

Section 5
P11, L23. At least in the conclusion, the statistical uncertainty should be clearly specified as 1-sigma standard deviation for a one day Langley plot. From Fig. 5, this was estimated to 0.3% (Mauna Loa) and 0.5% (Izana) . "…a single Langley plot will be typically within 1% of the mean". What exactly does the 1% signify? 95% confidence interval?

Following the changes in the statistical analysis, we have used now the 95% confidence interval, therefore 2-sigma. This has been clarified in the text: "Applying a 2-sigma criterion, the typical calibration uncertainty for a single Langley plot is ~0.7-0.9% (at the 95% confidence level). The necessary averaging of at least 10 Langley-derived extraterrestrial signals reduces the uncertainty to 0.25% at Mauna Loa and 0.4% at Izaña."

Why should the averaging be replaced by the temporal linear fit? Fitting a straight line would be a generalized method, including averaging as a special case with a line with slope zero.

The reference instruments are drifting by 0.1% per year. This is negligible for 2-3 week periods that are needed to accumulate 10-15 Langleys. Therefore averaging is a reasonable approach. If the deployment is very long (as in Figure 5), the linear fit is more adequate.

P11, L32. The discussion about the subtropical jet was not really conclusive.

The sentence has been softened to: "Furthermore, more investigations are needed to explore whether the subtropical jet above Izaña is a  possible explanation for the increase in the Langley plot residuals in this station during the spring months."

**Technical corrections**
There are different rules about capitalization, but I think in the context of e.g. "direct sun measurements", "sun photometry" etc., the common practice is to not capitalize "sun".
P1,L11. "…this uncertainty being smaller…"
P3, L15. "…it is the reference observatory…"
P4, L18. "…direct sun signal…"
P6, L2. "Smirnov"
P8, L9. "a certain", or even better "a slight"?
P11, L14. "…we find a climatological average…"
P12, L4. "signal losses"

OK, thanks.

[revised manuscript text omitted]

---

## Author Comment (AC2) · 20 Aug 2018

Dear editor,

Concerning the editor comment: "You need to strengthen more the "physical processes" part of the paper and make sure that the main overall conclusions are not limited to the accuracy and stability criteria. At this stage this could be achieved with some clear alignment of the focus of the article as this is reflected in the title, the abstract and the last paragraphs of the introduction and summary/conclusion sections. Please be prepared to elaborate further on the factors that affect the Langley plots validity and the estimation of the relevant uncertainties, during the 8 weeks provided for

the full review. "

We have elaborated the physics behind the Langley plot performance, by including several sentences throughout the text, as well as two paragraphs in sections 4.2 and 4.3 that describe in detail de processes that can change the atmospheric transmission during the Langley plot calibration: aerosols, pressure, ozone, airmass calculation, etc. Several references have been also added to reinforce the analysis.

Please also note that a new author, Alberto Berjón, has been added (it was missing by mistake in the first submission).

Best regards,

Carlos Toledano